# A Graph-Based Technique for Securing the Distributed Cyber-Physical System Infrastructure

**DOI:** 10.3390/s23218724

**Published:** 2023-10-26

**Authors:** Maxim Kalinin, Evgenii Zavadskii, Alexey Busygin

**Affiliations:** Institute of Computer Sciences and Cybersecurity, Peter the Great St. Petersburg Polytechnic University, 29 Polytekhnicheskaya ul., 195251 St. Petersburg, Russia

**Keywords:** adaptation, attack graph, cyber-physical system, functional dependencies graph, functional infrastructure, security, virtual isolated network

## Abstract

Spreading digitalization, flexibility, and autonomy of technological processes in cyber-physical systems entails high security risks corresponding to negative consequences of the destructive actions of adversaries. The paper proposes a comprehensive technique that represents a distributed functional cyber-physical system’s infrastructure as graphs: a functional dependencies graph and a potential attacks graph. Graph-based representation allows us to provide dynamic detection of the multiple compromised nodes in the functional infrastructure and adapt it to rolling intrusions. The experimental modeling with the proposed technique has demonstrated its effectiveness in the use cases of advanced persistent threats and ransomware.

## 1. Introduction

A cyber-physical system is an integrated digital and technical infrastructure responsible for monitoring and managing various technological processes that have different physical principles and properties. In the last few decades, due to the progressive development of advanced information and communication technologies, cyber-physical systems have undergone reasonable changes, marshalling distributed architecture, wireless interactions, and smart automation. The introduction of new software and hardware features that implement decision-making, machine-to-machine and machine-to-human integration has provided new opportunities for the digital production ecosystem, improving the efficiency and reliability of industrial control environments and business processes. But in this area, digitalization, flexibility, and autonomy increase responsiveness to emerging cybersecurity. The growing cybersecurity risks associated with multiple security impacts and sensitive consequences of malicious activity in cyber-physical distributed infrastructure are evidenced by different analytic agencies and world-leading companies. Symantec reports [1] that 65% of work teams around the world are facing advanced persistent threat (APT) attacks. Cisco estimates 40% of small and medium-sized industrial companies meet cybersecurity threats that interrupt their digital industry for at least 8 h [2,3].

Despite the development of advanced protection tools, the techniques used by intruders to downgrade or break cyber-physical systems are constantly progressing. This is why the APT attacks, in which intruders put to use various combinations of undeclared features, misconfiguration, zero-day vulnerabilities, and other security flaws in the functional units, cannot be detected by conventional network protection mechanisms, e.g., honeypot technique [4,5,6,7,8,9,10,11,12,13,14], intrusion detection/prevention and machine learning-based detectors [15,16,17,18,19,20,21,22,23,24,25,26,27], adaptive bioinspired methods [28,29,30,31,32,33]. Therefore, a novel approach to ensuring the security of cyber-physical systems is required, concerning the variability and complexity of cyberthreats as well as the flexibility and distribution of cyber-physical infrastructures.

To provide an asymmetrical response to cybersecurity threats in cyber-physical systems, a protective approach joining a pair of graph-based methods has been proposed:Active security sensing is based on an intelligent analysis of–A functional dependencies graph of the connected functional nodes;–A graph of potential attacks. This method provides step-by-step detection of intrusion rolling in the functional infrastructure of the monitored cyber-physical system;Dynamic counteraction to the detected intrusion is based on a predictive reconfiguration of the functional infrastructure of the monitored cyber-physical system. When the compromised nodes are eliminated from the system, the system’s functional process is rebuilt to restore the mission of the system and minimize the consequences of the destructive attacking actions.

Detecting destructive activity in the functional infrastructure of the cyber-physical system and downgrading negative effects, the proposed solution is part of the symmetry/asymmetry aspect meaningful to the reliable functioning of protected cyber-physical systems. To present the graph-based technique, the rest of the paper is organized as follows: Section 2 presents our technique proposed for securing the cyber-physical systems with the graph-based analysis; Section 3 shows the results of experimental modeling of our approach dealing with the APT attack and ransomware, demonstrating the effect of the proposed solution; Section 4 summarizes our work, comparing it to the related methods; and, finally, Section 5 concludes the work and sets the plan for further research.

## 2. Materials and Methods

Generally, to ensure the security of a cyber-physical infrastructure, the following sequence has to be implemented:Detect malicious actions and compromised nodes;Exclude the detected compromised nodes from the functional infrastructure;Re-build the functional path in the system in order to hold a technological process.

Constantly varying the attack technique, an adversary needs to perform active monitoring of the system undergoing the attack. Formally, nodes of the functional infrastructure of the cyber-physical system $V=\{v_{1},v_{2},\cdots ,v_{n}\}$ provide a set of functions $F_{v_{i}}=\left\{f_{1}^{\left(i\right)},f_{2}^{\left(i\right)},\cdots \right\}$, where $f_{k}^{\left(i\right)}$ denotes the functions supported by the node $v_{i}$, to ensure the correct flow of technological processes.

Functional nodes form a chain to realize certain technological processes: $R_{process_{j}}=\left\{v_{1}^{\left(j\right)},v_{2}^{\left(j\right)},\cdots ,v_{m}^{\left(j\right)}\;|\;f_{1}^{\left(j\right)}\in F_{v_{1}^{\left(j\right)}}\;\&\;\cdots \;\&\;f_{m}^{\left(j\right)}\;\in \;F_{v_{m}^{\left(j\right)}}\right\}.$ Such a network of connected functional nodes is called a functional dependencies graph. Figure 1 plots a sample of this graph for the cyber-physical infrastructure, some nodes of which are connected to a sequence implementing a technological process. A technological process is presented in the graph as a path formed by the nodes connected by arcs. For example, the cyber-physical system is presented by segments *A*, *B*, *C*, *D* and functional nodes 1, 2, 3, 4, 5, 6. Nodes of each segment have the similar functions. A technological process is plotted by black edges (this is a functional dependencies graph). Unconnected nodes do not participate in the functional process. To provide system security, one needs to keep the functional process due to cyberattacks influencing the functional process.

To catch a cyberattack rolling into the undergoing attack system, additional nodes, the indicator nodes ($v_{indicator}$), are introduced to the functional dependencies graph. They should be characterized with the following properties:Virtuality: they should be able to run and shut down online adequately to the complexity of the system;Indistinguishability: they have to be similar to the original (real) nodes placed in the given segment of the reflected cyber-physical system for the attacker’s side. The indicator nodes do not contain decoy objects and fully correspond to real nodes;They have to support a set of the same functions to implement the same technological processes virtually;They have to be fully controlled (e.g., their operating system) from outside by the defending facility.

Figure 2 is a revision of Figure 1, where the indicator nodes are inserted in every segment of the functional infrastructure. In Figure 2, light blue vertices are the indicator nodes and dark blue vertices are the real nodes of the cyber-physical system.

By conducting an intrusion into the functional infrastructure of the cyber-physical system, the attacker initiates a network interaction with some selected nodes in the attacked functional network. When contacting the indicator node, a malicious impact on the functional infrastructure is registered. It is worth noting that there is no regular network communication with indicator nodes from the attacker-controlled nodes. However, this is not a detecting factor, as this node may be involved in another process that does not use the compromised nodes. The worst attack scenario is one where no indicator node is affected and all compromised nodes keep working in the system’s functional infrastructure.

In order for this method to work, it is necessary to provide:
$$lim_{n\to \infty }\left(P\left(interaction\;with\;the\;v_{indicator}\right)\right)\to 1,$$ where *n* is the number of indicator nodes that have been built.

To determine the set of compromised nodes, one needs to set the condition of assigning the node $v_{i}$ to the set of compromised nodes $V_{compromised}$:

**Proof** **1.**A compromised node is a node that is included in the graph $G_{interaction}$ constructed using the backward path from the indicator node along the trace of node interactions: $v_{i}\in V_{compromised},\;if\;v\;\in \;V\left(G_{interaction}\right)$. □

Figure 3 shows the sample where six nodes are compromised in three segments. The attacker impacts the indicator node 7 in segment *B* (labeled: B/7). The attack entry point is A/1.

Applying Proof 1 to determine the compromised nodes, the following graph is obtained: the indicator node B/7→B/2; B/2→B/5; B/5→(B/4,C/3); C/3→C/1; C/1→(C/6,A/5); A/5→(A/2,A/1).

As a result, the sequence {A/1,A/2,A/5,C/1,C/3,C/6,B/5,B/4,B/2} is considered compromised. All nodes of $R_{process}$ have to be excluded from the system’s functional infrastructure.

Given the limited set of nodes that perform certain technological functions in the cyber-physical system, it is necessary to maximize the accuracy of identifying the compromised nodes. It can be accomplished with a potential attack graph. Each indicator node has to be associated with a potential attack graph that defines a trace from the entry point of the attack to this indicator node. Concerning this case, we have to state the following:

**Proof** **2.**A node is considered a compromised one if it is included in the graph resulting from the intersection of the potential attack graph $G_{attack}$ of the affected indicator nodes and the functional dependencies graph $G_{interaction}$, constructed according to Proof 1: $v_{i}\in V_{compromised},\;if\;v\;\in \;V\left(G_{interaction}\;\cap \;G_{attack}\right)$. □

If the indicator node interacts with a node that is not included in the potential attack graph, it has to be included in it. This provides at least one compromised node.

Let the potential attack graph for the indicator node B/7 include the nodes A/1,A/2,A/3,A/5,C/1,C/3,B/5,B/1,B/2,B/3. Following Proof 2, the sequence {A/1,A/2,A/5,C/1,C/3,B/5,B/2} is considered the compromised one. The nodes C/6 and B/4 can be used when performing re-arrangement of the functional path to keep a technological process running correctly.

Let us suppose several indicator nodes alarm us of a malicious influence; for example, the node D/1 is the attack entry point, and 3 indicator nodes are affected (Figure 4). Let these nodes correspond to the following attack graphs:B/7−A/1,A/2,A/3,A/5,C/1,C/3,B/5,B/1,B/2,B/3;D/5−A/1,A/2,A/5,C/1,D/1,D/3;D/7−D/1,D/3,D/2.

The graph $G_{interaction}$ includes the nodes A/2,A/5,C/1,C/3,C/6,B/5,B/2,B/4,D/1. As this attack affects 3 indicator nodes, modification of Proof 2 is required:

**Proof** **2a.**A node is considered a compromised one if it is included in the graph resulting from the intersection of the graph, which is a union of the potential attack graphs $G_{attack}$ of the appropriate affected indicator nodes and the functional dependencies graph $G_{interaction}$, constructed according to Proof 1: $v_{i}\in V_{compromised},\;if\;v\;\in \;V\left(G_{interaction}\;\cap \;\left(G_{attack}^{1}\cup G_{attack}^{2}\cup \cdots \right)\right)$. □

According to Proof 2a, in our example of system representation, the sequence {A/2,A/5,C/1,C/3,B/5,B/2,D/1} is compromised. Nodes A/2 and A/5 have to be excluded from the system’s functional infrastructure, but they are not compromised. To solve this issue and preserve these nodes for the system, Proof 3 has been introduced:

**Proof** **3.**A node is not a compromised one if the indicator nodes do not signal malicious activity on it, and no part of the functional dependencies trace leads to the indicator nodes affected by the attack. $v_{i}\notin V_{compromised},\;if\left(\forall \;v_{indicator}^{\left(j\right)}:is\_signal\left(v_{indicator}^{\left(j\right)}\right)=false\right)\;\&\;\left(v_{i}\notin s_{km}=<v_{indicator}^{\left(k\right)},\cdots ,v_{indicator}^{\left(m\right)}>\forall \;v_{indicator}^{\left(k\right)},v_{indicator}^{\left(m\right)}:is\_signal\left(v_{indicator}^{\left(k\right)}\right)=true,is\_signal\left(v_{indicator}^{\left(m\right)}\right)=true\right),$ where is_signal—the function that returns the status of the indicator node. □

For the limits of the paper, simultaneous attacks on a cyber-physical system are not considered.

For Proof 3, the nodes A/2 and A/5 can be excluded from the sequence {A/2,A/5,C/1,C/3,B/5,B/2,D/1} derived from Proof 2a. Therefore, only the compromised nodes are eliminated from the functional infrastructure.

Suppose the node A/5 is compromised, and it is not interacting with the indicator nodes. Then there will be a compromised node in the functional infrastructure that can keep performing destructive activity. The detection of such nodes is possible through the detection of behavioral signatures of compromised nodes. To explore such signatures, all compromised nodes identified in the first phase should be separated; they are to be located in an isolated virtual network. If doing so, this network must be similar to the original one (Figure 5).

The compromised nodes in the isolated virtual network continue performing destructive actions, the analysis of which allows one to identify signatures of their malicious behavior. Based on the collected signatures, one can repeatedly check the infrastructure nodes that are still working after the exclusion of the compromised nodes. If the infrastructure nodes meet signatures, they should be also excluded from the functional infrastructure (Figure 6).

At each stage of this algorithm, the implementation of the technological process is rebuilt to ensure its continuous operation.

Figure 5 shows an example of this restructuring—process implementation $R_{process}$ has been changed to $R_{process}^{{}^{\prime }}$: $R_{process}=\left\{v_{A/2},v_{A/5},v_{C/1},v_{C/3},v_{C/6},v_{B/5},v_{B/2},v_{B/4}\;|\;f_{1}\in F_{v_{A/2}}\;\&\;\cdots \;\&\;f_{8}\;\in \;F_{v_{B/4}}\right\}\to R_{process}^{{}^{\prime }}=\left\{v_{A/2},v_{A/4},v_{C/2},v_{C/4},v_{C/6},v_{B/6},v_{B/1},v_{B/4}\;|\;f_{1}\in F_{v_{A/2}}\;\&\;\cdots \;\&\;f_{8}\;\in \;F_{v_{B/4}}\right\}$

To evaluate the proposed technique, the experimental study has been arranged.

## 3. Results

The experimental study has been aimed at testing the proposed solution for its resistance to different scenarios of cyberthreats. Two of the most crucial threats affecting cyber-physical systems today were chosen for modeling: the APT attack and ransomware (e.g., WannaCry).

For testing, we developed a modeling simulator that models the intrusion stages based on a given configuration of the cyber-physical system and generates a graph of the dependence of the number of compromised network nodes on time in the case of the use of the proposed solution and in the case of the absence of it.

The modeling simulator includes three separate entities:An entity of the cyber-physical system’s node, each instance of which has a unique identifier, supports a certain set of functions and can be an indicator node;The entity of the attacker. His task is to perform malicious actions against instances of the CPS node entity as part of the attack according to a predefined scenario or based on a random selection of the target node. A compromised node can be used as a proxy node to redirect malicious actions to another node or as a standalone attacker node;The entity of the cyber-physical system’s supervisor provides the functioning of the proposed complex solution by receiving messages from indicator nodes, determining the set of compromised nodes according to the introduced proofs, disconnecting them from the simulated cyber-physical system’s network, and rebuilding the set process $R_{process}$. The isolation of compromised nodes into an isolated network and the collection of their behavioral signatures have been replaced by a delay of 20 s.

Each simulator’s entity is a separate process that interacts through bi-directional message queues. The simulated cyber-physical system contains 20 regular nodes and 16 indicator nodes. Simulation of the system graph was conducted on the workstation with Intel Core i7-4770HQ CPU and 16GB RAM.

The APT attack was simulated according to the following restrictions:Manual mode attack—the attacker can interact with only one node at time;Attack on each node includes a compromise and anchoring phase in 12 s and a download and launch phase of the necessary tools to select the next target node in 52 s;Interaction with the cyber-physical system’s nodes only through one entry node;Interaction with the internal target node requires a network link from the input node to the target node.

Figure 7 represents one of the scenarios of the APT attack that was modeled in the simulator. The attack entry node is D/3. During the invasion, the nodes D/1,D/2,C/1,C/3,A/5 are compromised sequentially. In the case of the proposed approach, the intruder has affected the node D/7 after compromising the node C/3 (Figure 8).

The node D/7 is an indicator node; it has detected the attack. The graph of potential attacks for it includes nodes D/1,D/2 and D/3. According to the previously stated algorithm, nodes D/3,D/1,D/2 are located in an isolated virtual network. Since the entry node was removed from the protected infrastructure, the remaining compromised nodes are inaccessible to the attacker, and propagation to other nodes is impossible.

After performing a behavioral analysis of the compromised nodes located in an isolated network, signs of compromise can be identified, and the remaining two nodes C/1 and C/3 are removed from the functional infrastructure.

Ransomware uses malware that selects target hosts and compromises them automatically. Figure 9 shows a sample graph for this test case.

The attack entry node is A/2. Nodes A/1,A/3,A/5,A/6 and A/7 have been defined as targets. The latter nodes are indicator nodes; hence, the malicious impact is detected. The potential attack graph for these indicator nodes includes the nodes A/1,A/2,A/3 and A/1,A/2,A/3,C/2,D/2,D/1, respectively. According to the above proofs, the following set of nodes has been identified: A/1,A/2,A/3. These nodes are moved to an isolated virtual network for behavioral analysis. As the node A/5 was also compromised, it continued to perform malicious activity, infecting new C/1,A/4,A/8,A/9 nodes. After identifying the signatures of the compromised nodes’ behavior in the isolated network, the compromised nodes A/5,A/4 and C/1 were removed from the protected network. Thus, the encrypting attack is localized, and its further development is prevented.

Protection effectiveness has been measured as a symmetry of response, namely how many compromised nodes are correctly detected and eliminated from the system. For example, Figure 10 and Figure 11 demonstrate a symmetric response to an APT attack—100% of the compromised nodes were detected by the proposed technique.

After allocation of the compromised nodes in the isolated network, some scenarios can finish with an $R_{process}$ recovery error due to the unavailability of the nodes that can provide a required function.

During the experiments, it was also figured out that the detection rate depends on the number of involved indicator nodes, as the probability of the attacker impact reasonably depends on the sensor field density.

## 4. Discussion

Plenty of advanced protection techniques are designed to counteract intruders in the cyber-physical networks. Honeypot technique, intrusion detection/prevention and machine learning-based detectors, bioinspired and immune methods are among them. Table 1 summarizes the proposed technique with the related methods.

As a cyber-physical system can be represented as a functional infrastructure of the connected nodes, one of the prevalent conventional protection techniques is a honeypot that diverts an attacker from the crucial assets [4]. Honeypot hosts must be indistinguishable from nodes in the real network to attract and deceive adversaries. The majority of the honeypot systems applied for protection of cyber-physical systems are low-interactive [5,6,7,8,9,10,11], and this simplifies discovering and by-passing the honeypots by attackers.

Unlike a honeypot, deception technology operates within the standard algorithm of cyberattack: collecting and analyzing data, selecting target nodes, and performing lateral propagation [12]. The attacker’s distraction is performed by introducing decoy nodes that host false parameters. If they are detected, the attacker can apply them to laterally propagate to the corresponding specially created decoy nodes. Game theory is commonly applied to manage deception systems [13,14]. Unfortunately, all the competitive problems cannot be analyzed with the help of game theory, and each player has to know the cost function of the other one.

In refs. [15,16,17,18], the researchers introduce the machine learning techniques to detect intrusions and anomalies in the behavior of the cyber-physical systems, demonstrating the high accuracy of the prototyped detectors. Artificial swarm algorithms in the IDS systems can achieve even higher quality in detecting attacks based on distributed knowledge-driven analysis of the inter-node network. In Ref. [19], an intelligent agent-based ensuring security is specified.

Refs. [20,21,22,23,24,25,26,27] suggest the security optimization method based on a symbiosis of convolutional neural networks and genetic algorithms. High accuracy of detecting cyberattacks is achieved for typical attack datasets. However, in cases of attacks not foreseen in the training models and in the absence of a correlation of trait values, the compromised nodes cannot be detected by the machine learning. It also should be taken into account that inter-node interactions are encrypted on the transport layer to protect data transmission, and this reduces the effectiveness of the machine learning methods.

The authors of the work [26] proposed to use the theory of complex networks for dynamic assessment of network vulnerability, which can be used to detect the rolling attacks and predict the direction of intrusion propagation. However, the authors of that article note that the implementation of their method has a number of limitations, such as the availability of real-time and historical data, uncertainties in vulnerability assessment models, and accuracy of collected data.

Also, honeypot/deception and intellectual methods allow for the alarming of malicious activity only, but do not counteract a security threat itself. These methods have been performed so far manually, which require significant resources given the increasing scale and greater autonomy of cyber-physical systems. To make the duty easier, refs. [28,29,30,31,32,33] have proposed a number of methods grounded in a bioinspired basis and control theory to minimize the impact of destructive actions by attackers. The correct reaction of such systems has a probabilistic nature. For example, a concept of an immunization system has been designed to solve the same task. In ref. [33], the immunization system detects an intrusion in a controlled system and supplies a symmetrical cure against specific attacks on critical nodes. All this group of solutions is truly applied to the ’a posterior’ defense of cyber-physical systems, but they still do not provide the ability to prevent the further spreading of the attack over the injured system, because they do not identify the attack’s source node, the compromised nodes, that remain performing destruction in the attacked infrastructure.

Within our research, a cryptographic protection (e.g., refs. [34,35]) is not considered because it does not solve the problem. Communications between nodes can be protected by encryption, but the technological functions of the system fail due to specific cyberthreats [36].

As intruders utilize undeclared features, misconfigurations, zero-day vulnerabilities, and system security flaws, most of cyberattacks cannot be detected by conventional network protection mechanisms [37]. Hence, based on the presented research background, our approach is aimed at elaborating a novel technique for the protection of a functional network of cyber-physical systems. The proposed graph-based technique is at first based on the method of active analysis of the functional dependencies graph, which allows us to identify compromised nodes rapidly while an attack is in progress. The objective of the second proposed method is to counteract the detected intrusion by applying predictive reconfiguration of the system’s functional infrastructure. To do it, this method analyzes a potential attack graph and moves the compromised nodes to the isolated virtual network to analyze their behavior in detail, then does a work on errors, and, finally, re-configures the system technological process in the infrastructure of nodes engaging the unaffected nodes.

The related works listed in Table 1 do not provide a comprehensive technique for detecting the malicious impact of an attacker, minimizing its consequences through the identification of the compromised nodes and their disconnection from the system’s functional infrastructure. The proposed solution allows identifying the compromised nodes with high accuracy due to the stated conditions and a mechanism for collecting behavioral signatures of data nodes in an isolated virtual network to identify previously undetected nodes. Experimental modeling of our solution on APT attack and ransomware sample scenarios has demonstrated correct work and an asymmetric response to cyberthreats of different kinds and variability.

For effective use of the proposed method, it is necessary to ensure a fast and seamless allocation of the compromised nodes into an isolated virtual network similar (from the attacker’s point of view) to the protected network. Creating this virtual network requires significant computing resources. However, it allows identifying the behavioral signatures of the compromised nodes and supplementing their databases, which can then be used by the IDS [38].

It is necessary to be concerned that if some nodes are excluded from the network, the technological process may not be recoverable due to the absence of nodes that perform the required industrial functions. It follows that mechanisms for redundancy of the most critical nodes and recovery of compromised nodes are required. Identification of critical nodes can be performed using the graph neural network-based method proposed in [39]. In this case, the backup nodes should be in an off state or located in a separate isolated network to prevent their compromise.

The implementation of the second mechanism involves restoring a compromised node from a backup copy and embedding it into the network infrastructure of the cyber-physical system.

## 5. Conclusions

Ensuring the cybersecurity of modern self-organizing systems (the Internet of Things, machine-to-machine networks, software-defined networks, cyber-physical systems, etc.) is a crucial challenge, as successful attacks on the components of critical infrastructure may lead to disruption of industrial processes, destruction of expensive devices, and breaking individual, social, and economic life. The goal of our research was to propose a novel approach to online detection and active counteraction to cyber-intrusions and thus enhance the security of a functional distributed infrastructure of critical systems.

The designed methodology joins two methods: (1) analysis of the functional dependencies graph, which allows us to identify the system compromised nodes during the attack in progress; and (2) countering intrusions using the reconfiguration of the system’s functional infrastructure, which analyzes the attack graph and moves the compromised nodes to an isolated virtual network so as to analyze their behavior, perform the ’work-on-errors’, and re-build the technological process with the unaffected nodes.

Experimental modeling has demonstrated the correct work of the proposed methodology and the asymmetric response of our proposed approach to the APT and ransomware scenarios.

Our further work is targeted at the use-case application of our method to the definite vehicle ad hoc network (VANET), ensuring its adaptive and resistible protection.

## Figures and Tables

**Figure 1 sensors-23-08724-f001:**
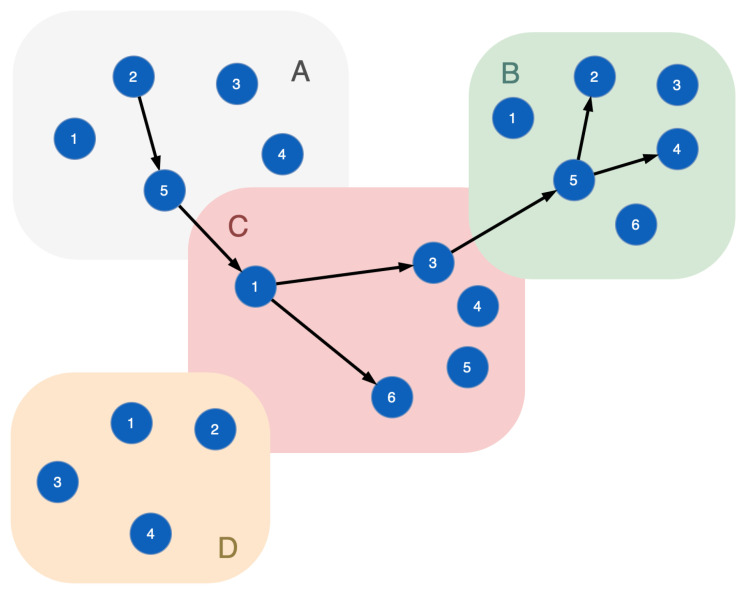
A sample of the functional dependencies graph: black edges—connections of the functional nodes that implement a certain technological process.

**Figure 2 sensors-23-08724-f002:**
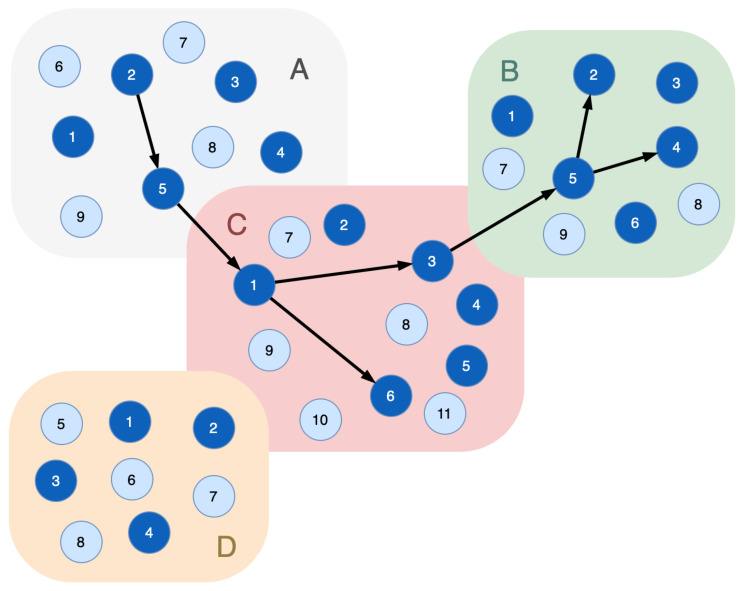
A sample of the functional dependencies graph with indicator nodes: light blue vertices—the indicator nodes; dark blue vertices—the real nodes of the cyber-physical system; black edges—connections of the functional nodes that implement certain technological process.

**Figure 3 sensors-23-08724-f003:**
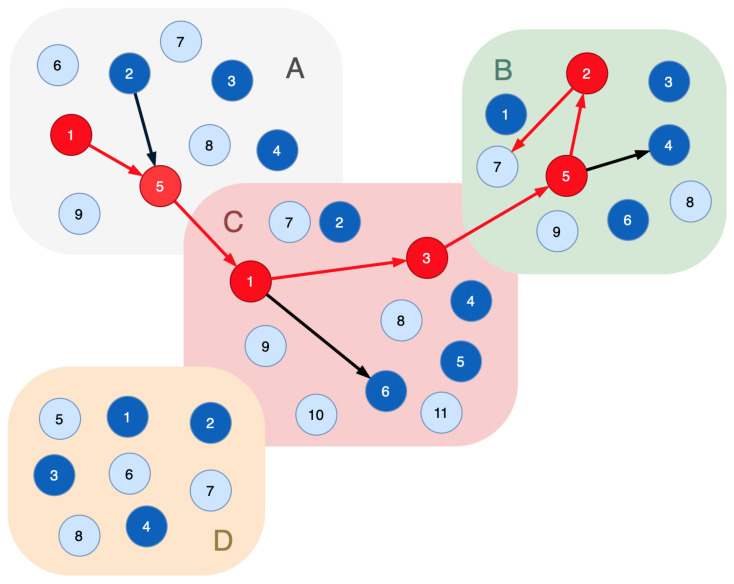
A sample of the functional dependencies graph with six compromised nodes: light blue vertices—the indicator nodes; dark blue vertices—the real nodes of the cyber-physical system; red vertices—the compromised nodes; black edges—connections of the functional nodes that implement a certain technological process; red edges—the movements of the intruder.

**Figure 4 sensors-23-08724-f004:**
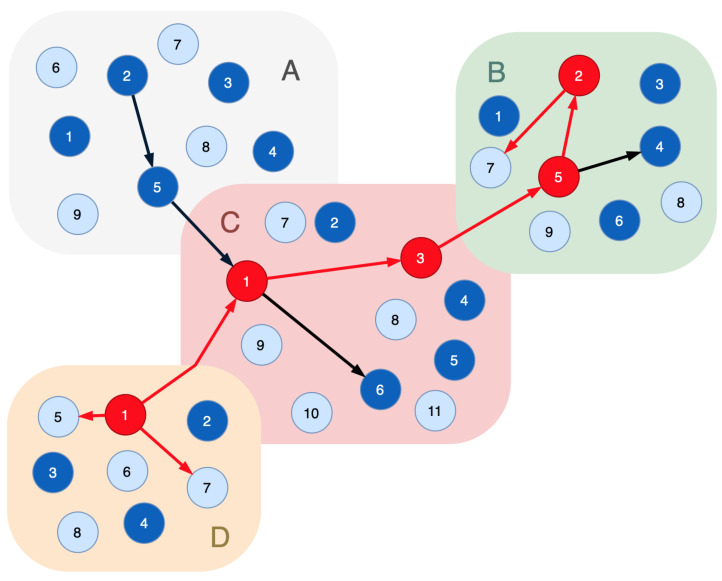
A sample of the functional dependencies graph with five compromised nodes: light blue vertices—the indicator nodes; dark blue vertices—the real nodes of the cyber-physical system; red vertices—the compromised nodes; black edges—connections of the functional nodes that implement certain technological process; red edges—the movements of the intruder.

**Figure 5 sensors-23-08724-f005:**
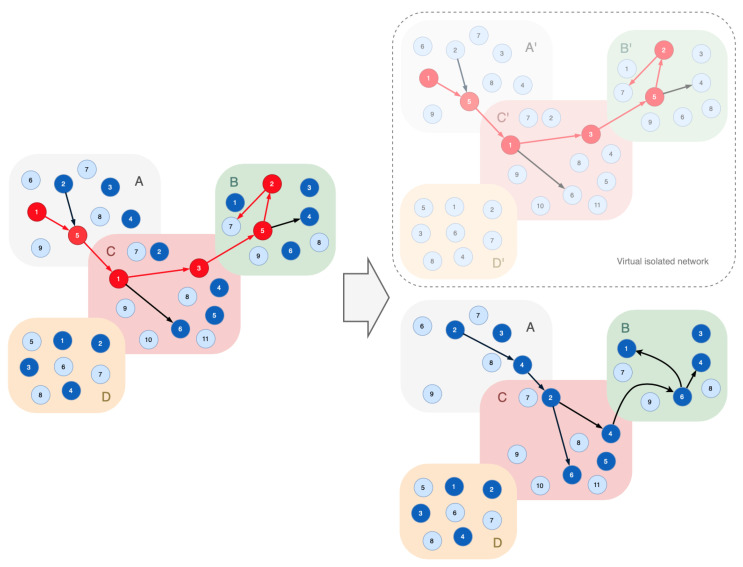
Restructured cyber-physical infrastructure.

**Figure 6 sensors-23-08724-f006:**
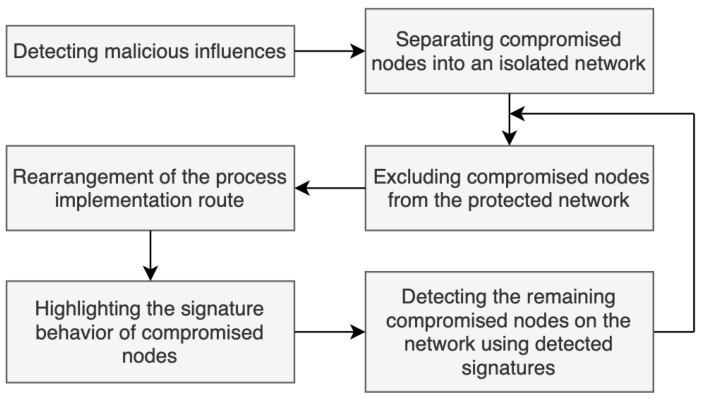
An algorithm for detection and exclusion of compromised nodes.

**Figure 7 sensors-23-08724-f007:**
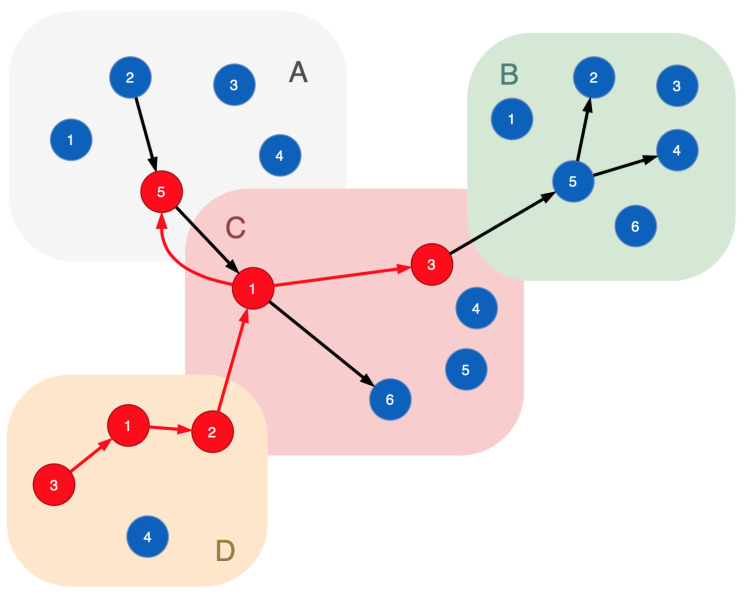
Functional dependencies graphs without using the proposed solution (case of APT attack).

**Figure 8 sensors-23-08724-f008:**
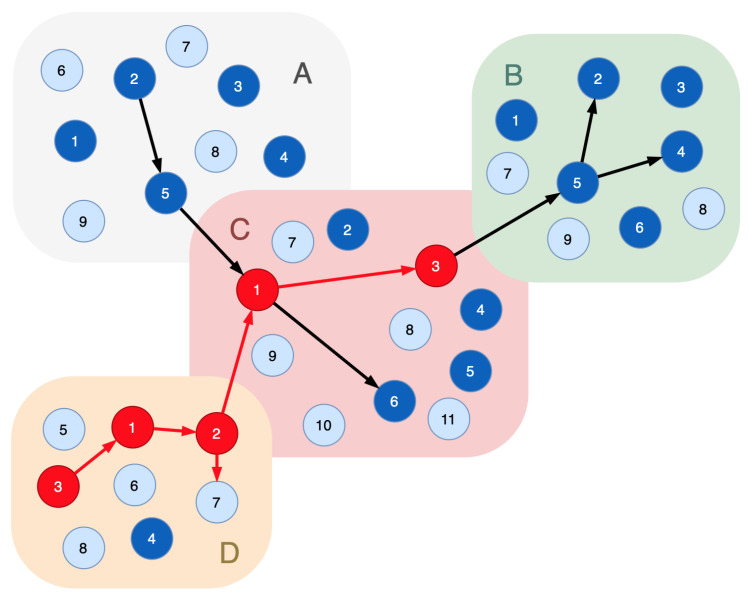
Functional dependency graphs with using the proposed solution (case of APT attack).

**Figure 9 sensors-23-08724-f009:**
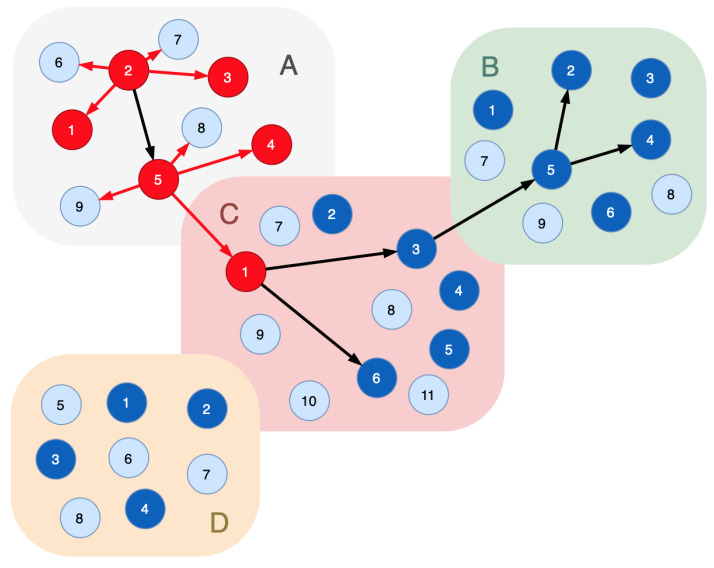
Functional dependencies graph for ransomware case.

**Figure 10 sensors-23-08724-f010:**
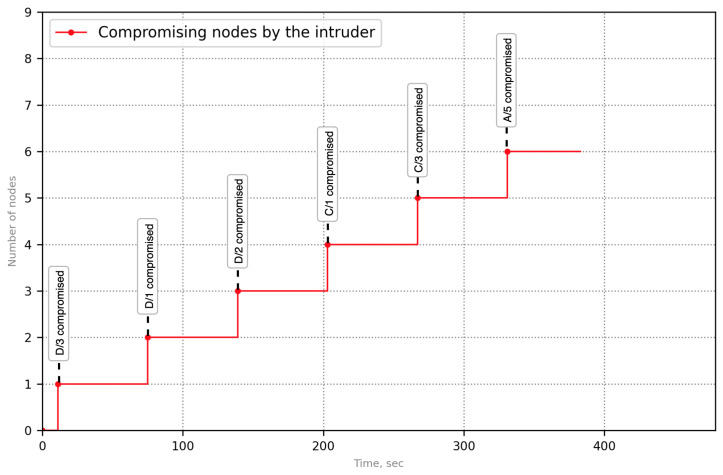
Node compromise while the APT attack rolls without the proposed solution.

**Figure 11 sensors-23-08724-f011:**
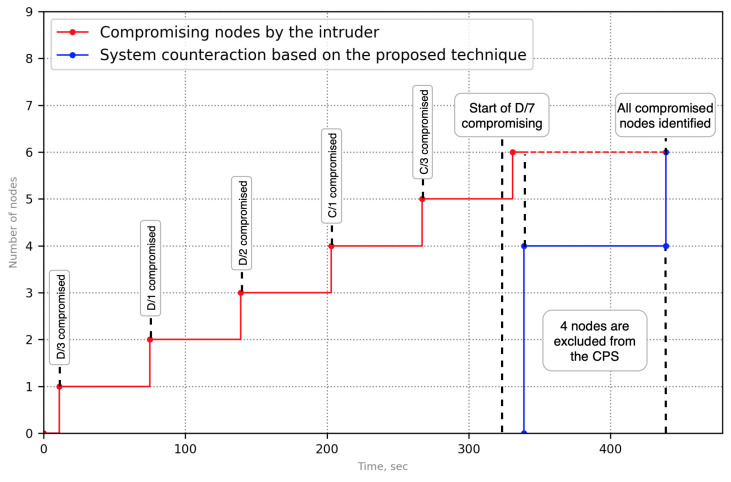
Node compromise while the APT attack rolls with the proposed solution.

**Table 1 sensors-23-08724-t001:** Comparison of the proposed solution with the related works.

Methods	Detection of the Attacks Unintended by Configuration	Standalone Attack Countermeasures	Automatic Update of the Signature Database	Demand for the Computing Resources
**Honeypot/Deception** [4,5,6,7,8,9,10,11,12,13,14]	+ ^1^	—	—	—
**IDS/ML** [15,16,17,18,19,20,21,22,23,24,25,26,27]	—	—	—	—
**Bioinspired/Immune methods** [28,29,30,31,32,33]	+	+	—	—
**Proposed technique**	+	+	+	+

^1^ ’+’—feature is provided by method; ’—’—feature is not provided by method. Footnote under the table is added to explain plus and minus signes.

## Data Availability

Not applicable.

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
