# Peer review of "A Graph-Based Technique for Securing the Distributed Cyber-Physical System Infrastructure"

_sensors, 2023, doi:10.3390/s23218724_

Round 1
Reviewer 1 Report
The paper proposes a comprehensive technique that represents a distributed cyber-physical system infrastructure by graphs. The following problems still exist, and the author is suggested to revise them carefully.
1. In Figure 1, Why do some nodes have no links? What do the nodes with different colors in Figure 2 represent? Please provide a detailed description and explanation of Figure 1 and Figure 2.
2. Does the method proposed in this article have an impact on the system delay? Please provide a detailed explanation.
3. The research on the current status of literature in this article is not sufficient, and some research results are relatively old. Suggest the author to add some latest research on data security or distributed cyber-physical system.
[1] Kou, L.; Wu, J.; Zhang, F.; Ji, P.; Ke, W.; Wan, J.; Liu, H.; Li, Y.; Yuan Q. Image encryption for Offshore wind power based on 2D-LCLM and Zhou Yi Eight Trigrams. International Journal of Bio-Inspired Computation, 2023. https://doi.org/10.1504/IJBIC.2023.10057325
[2] Alotaibi, B. A Survey on Industrial Internet of Things Security: Requirements, Attacks, AI-Based Solutions, and Edge Computing Opportunities. Sensors 2023, 23, 7470. https://doi.org/10.3390/s23177470
[3] Ünal ÇavuÅŸoÄŸlu and Abdullah Hulusi Kökçam. A new approach to design S-box generation algorithm based on genetic algorithm. International Journal of Bio-Inspired Computation, 2021, 17, 1. pp 52-62. https://doi.org/10.1504/IJBIC.2021.113360
[4] Zang, T.; Wang, Z.; Wei, X.; Zhou, Y.; Wu, J.; Zhou, B. Current Status and Perspective of Vulnerability Assessment of Cyber-Physical Power Systems Based on Complex Network Theory. Energies 2023, 16, 6509. https://doi.org/10.3390/en16186509
4. Has the author considered the network nodes in practical applications? Can you provide a case study.
5. Table 1 compares the proposed approach with the related works, suggest enhancing the analysis of the results of Table 1.
Author Response
Thank you for your work with our manuscript and for your email enclosing the reviewers’ comments. We have carefully reviewed the comments and have revised the manuscript accordingly. Our responses are given in a point-by-point below.
We uploaded the updated submission. Changes to the manuscript are highlighted in the text. We hope the revised version is now suitable for publication and look forward to hearing from you in due course.
We have addressed the Reviewer#1 comments as follows:
Comment #1: In Figure 1, Why do some nodes have no links? What do the nodes with different colors in Figure 2 represent? Please provide a detailed description and explanation of Figure 1 and Figure 2.
Answer: We gratefully appreciate for you comment. The connected nodes implement a technological process. If the nodes do not implement the process, they are not linked on the graph. Different nodes have different colors: light blue vertices are the indicator nodes; dark blue vertices are the real nodes of the cyber-physical system. A detailed description is presented in lines 73-75: “Figure 1 plots a sample of this graph for the cyber-physical infrastructure, some nodes of which are connected to a sequence implementing a technological process; the nodes of each segment A, B, C, and D have the similar functions.” Legend of node colors is presented in the Figure 2 title: “… light blue vertices – the indicator nodes; dark blue vertices – the real nodes of the cyber-physical system; black edges – connections of the functional nodes that implement certain technological process”. We hope that this specification is enough for understanding the graph in the presented Figure.
Comment #2: Does the method proposed in this article have an impact on the system delay? Please provide a detailed explanation.
Answer: The proposed anti-attack method really affects the delays of the cyber-physical system. The impact factor is the need to rebuild the chain of nodes that implement the technological process. However, in our research, the algorithm that performs the rebuilding operation is not considered, because it depends on a concrete cyber-physical system topology and technological functions implemented in it. Additional nodes of the anti-attack system that are embedded in the cyber-physical system are virtual machines and run on a dedicated server. An isolated network implemented to collect indicators of compromise is also hosted on this server. Therefore, there is no other impact on the system delay.
Comment #3: The research on the current status of literature in this article is not sufficient, and some research results are relatively old. Suggest the author to add some latest research on data security or distributed cyber-physical system.
[1] Kou, L.; Wu, J.; Zhang, F.; Ji, P.; Ke, W.; Wan, J.; Liu, H.; Li, Y.; Yuan Q. Image encryption for Offshore wind power based on 2D-LCLM and Zhou Yi Eight Trigrams. International Journal of Bio-Inspired Computation, 2023. https://doi.org/10.1504/IJBIC.2023.10057325
[2] Alotaibi, B. A Survey on Industrial Internet of Things Security: Requirements, Attacks, AI-Based Solutions, and Edge Computing Opportunities. Sensors 2023, 23, 7470. https://doi.org/10.3390/s23177470
[3] Ünal ÇavuÅŸoÄŸlu and Abdullah Hulusi Kökçam. A new approach to design S-box generation algorithm based on genetic algorithm. International Journal of Bio-Inspired Computation, 2021, 17, 1. pp 52-62. https://doi.org/10.1504/IJBIC.2021.113360
[4] Zang, T.; Wang, Z.; Wei, X.; Zhou, Y.; Wu, J.; Zhou, B. Current Status and Perspective of Vulnerability Assessment of Cyber-Physical Power Systems Based on Complex Network Theory. Energies 2023, 16, 6509. https://doi.org/10.3390/en16186509
Answer: We gratefully appreciate for you comment. Our goal of the related works analysis was to specify the main directions in protection mechanisms applied for the functional network protection. Thanks for the list of contemporary articles you suggested. We have carefully studied the articles. The first and third articles describe the encryption algorithms. Within our research, a cryptographic protection is not considered because it does not solve the problem (for instance, the connections can be protected by encryption, but the technological function of the system is not executed due to DoS attack). The second article provides an overview of the current state of security in the industrial IoT and lists the methods that have been already discussed in our article in the “Discussion” section. The fourth article is devoted to vulnerability assessment and the application of this method to the security of a cyber-physical system. This method was added to the related works. Also, the latest articles were added to the bibliography to extend the field of contemporary related works.
Comment #4: Has the author considered the network nodes in practical applications? Can you provide a case study.
Answer: This research did not consider specific applications of the system functional nodes, such as HMI panels, software logic controllers, etc. We were targeted to develop a universal conceptual approach that can protect the cyber-physical system and does not depend on the system specific.
Comment #5: Table 1 compares the proposed approach with the related works, suggest enhancing the analysis of the results of Table 1.
Answer: We gratefully appreciate for you comment. The “Discussion” section provides an analysis of the related works aimed at detecting and countering attacks. Also, in this section, the shortcomings that can be solved by implementing the proposed method are pointed out. Table 1 summarizes exactly the specification presented in the “Discussion” section.
Reviewer 2 Report
English needs some improvement. This is throughout the submission, and it distracts from the paper.
The introduction was well done overall, and introduced the two graph-based methods that were used in the work. An additional discussion of the novelty of these would have been helpful for a reader.
The background aspects of this work are largely absent. What is the underlying need for this? Why are Cyber-Physical Systems unique? There is modern research that should be referenced in this work, but has not been. Areas such as attack graphs, network sensing, dynamic network restructuring and self-healing, moving target defense, active security and intelligent security, are all active and have significant research underpinning them.
The concept is interesting, and the implementation appeared well done. However, the research in this needed more depth, and greater links to the current state of research.
The references present are good, but there needs to be far more evidence of research. This publication needs a greater number of references, and also more modern ones.
There are grammatical errors throughout, and also many terms are strangely used. This distracts from the work. Spelling is fine.
For example, the title: "The graph-based technique for securing the distributed cyber-physical system infrastructure” would be better expressed as “A graph-based technique for securing distributed cyber-physical system infrastructure”.
Similar examples are across the entire paper.
Author Response
Thank you for your work with our manuscript and for your email enclosing the reviewers’ comments. We have carefully reviewed the comments and have revised the manuscript accordingly. Our responses are given in a point-by-point below.
We uploaded the updated submission. Changes to the manuscript are highlighted in the text. We hope the revised version is now suitable for publication and look forward to hearing from you in due course.
We have addressed the Reviewer's comments as follows:
Comment #1: English needs some improvement. This is throughout the submission, and it distracts from the paper.
Answer: Thank you for your valuable comment. We have revised English and did the proofreading. We hope that English became correct and easy to read.
Comment #2: The background aspects of this work are largely absent. What is the underlying need for this? Why are Cyber-Physical Systems unique? There is modern research that should be referenced in this work, but has not been. Areas such as attack graphs, network sensing, dynamic network restructuring and self-healing, moving target defense, active security and intelligent security, are all active and have significant research underpinning them. The concept is interesting, and the implementation appeared well done. However, the research in this needed more depth, and greater links to the current state of research.
Answer: We thank you for reminding us of this important point. The research background is indeed really large, but the merit of the proposed solution is a new concept of attack counteracting in the functional structure of the CPS. The differences of our solution comparing it to the alternatives in the same research domain are presented in the “Discussion” section and summarized in Table 1.
Cyber-Physical Systems are unique for our research because it is a functional adaptive (reconfigurable) network of functional nodes. And there is a set of new methods that treat the security of CPS through its topological structure (e.g., https://lpasquale.github.io/papers/ICSEDemo15.pdf).
Links to the current state of research are presented in the Discussion section, because we followed the “Instructions for Author”, namely the “Research Manuscript Sections” requirements. The manuscript contains the following sections: “Introduction”, “Materials and Methods”, “Results”, “Discussion”, “Conclusions”. In the “Discussion” section we observed the current trends of the adaptive CPS protection that are relative to our work.
Comment #3: The references present are good, but there needs to be far more evidence of research. This publication needs a greater number of references, and also more modern ones.
Answer: In the “Discussion” section we observed the current trends of the adaptive CPS protection that are relative to our work. The latest articles were added to the bibliography to extend the field of contemporary related works.
Comment #4: Comments on the Quality of English Language
There are grammatical errors throughout, and also many terms are strangely used. This distracts from the work. Spelling is fine.
For example, the title: "The graph-based technique for securing the distributed cyber-physical system infrastructure” would be better expressed as “A graph-based technique for securing distributed cyber-physical system infrastructure”.
Answer: We have revised English and did the proofreading. We hope that English became correct and easy to read. Also we have revised the title of the paper to follow your valuable recommendation.
Comment #5: Similar examples are across the entire paper.
Answer: We used the same graph to demonstrate our approach step-by-step. Fig. 1 plots the basic system, Fig. 2 plots the functional dependencies graph with indicator nodes; etc. Figures with a single throughout example illustrate the developments of the statements introduced in our research. We believe that this end-to-end example makes understanding of our approach easier.
Round 2
Reviewer 1 Report
Accept in present form